# Protection of LPS-induced intestinal injury in goslings by polysaccharide of *Atractylodes macrocephala* Koidz based on 16S rRNA and metabolomics analysis

Wanyan Li[1] ☯, Baili Lu[1] ☯, Shirou Pan[1], Shuaifei Bai[1], Jiayu He[1], Bingxin Li[1], Nan Cao[1], Xinliang Fu[1], Junhao Wei [2], Ying Chen[3], Yunmao Huang[1], Yunbo Tian[1]*, Danning Xu [1]*

**1** College of Animal Science and Technology, Zhongkai University of Agriculture and Engineering, Guangzhou, China, **2** Faculty of Applied Sciences, Macao Polytechnic University, Macao, China, **3** Department of Health Management, School of Health Economics and Management, Nanjing University of Chinese Medicine, Nanjing, China

☯ These authors contributed equally to this work.

* xdanning212@126.com

**Data availability statement:** All relevant data are within the manuscript and its Supporting Information files. https://doi.org/10.6084/m9.figshare.28616936.v1

## Abstract

The protective effects of the polysaccharide of *Atractylodes macrocephala* Koidz (PAMK) against lipopolysaccharide (LPS)-induced intestinal injury in goslings was determined using 16S rRNA analysis of cecal contents and serum metabolomics analysis. PAMK was administered to goslings following LPS-induced intestinal injury, and its effects were assessed. PAMK significantly reduced the serum levels of inflammatory factors including interleukin (IL)-6 and C-reactive protein and decreased the expression of pro-inflammatory cytokines including interleukin(IL)-1$\beta$, IL-6, and toll-like receptor 2 in jejunal tissues. Moreover, PAMK significantly upregulated the relative mRNA expression levels of the tight junction proteins *Zonula occludens*-1, *Occludin*, *Claudin*, and *Mucin*-2, enhancing the integrity of the intestinal barrier and alleviating LPS-induced intestinal injury. 16S rRNA sequencing revealed that PAMK could alleviate LPS-induced disruption of the intestinal microbiota structure and improve microbial diversity. Metabolomics analysis revealed that PAMK could influence key metabolic pathways, including the mTOR, PI3K-Akt, and FoxO signaling pathways, and regulate metabolites such as L-aspartic acid and S-adenosylmethionine. Integrated analysis indicated that PAMK could promote the enrichment of beneficial bacteria (e.g., *Allobaculum* and *Peptococcus*) while alleviating LPS-induced microbial dysbiosis by modulating the correlation between key metabolites and specific microbial populations. Overall, PAMK could alleviate LPS induced intestinal injury by enhancing intestinal barrier function, optimizing gut microbiota composition, and regulating metabolic signaling pathways. Our findings provide a novel strategy for maintaining the intestinal health of poultry and preventing intestinal diseases.

**Funding:** The National Natural Science Foundation, grant number 32102747 and 32202764; the Science Technology Planning Project of Guangzhou, grant number 2023A04J0741 and 2023E04J0022, the Special Projects in Key Areas of General Universities in Guangdong Province , grant number 2022ZDZX4022.

**Competing interests:** The authors have declared that no competing interests exist.

# 1 Introduction

Environmental factors contribute to the proliferation of harmful bacteria in water sources during goose rearing [1]. Goslings are frequently exposed to large quantities of gram-negative bacteria, leading to the accumulation of external toxins and bacterial endotoxins in their intestines. This accumulation often results in disease outbreaks, significantly impacting production and thereby affecting the economic benefits of poultry farmers.

LPS, a key virulence factor of gram-negative bacteria, exhibits strong immunogenicity and toxicity [2].Upon accumulation in the body, LPS triggers intestinal inflammation, disrupts microbial homeostasis, and causes metabolic disorders. LPS damages intestinal epithelial cells by impairing nutrient absorption, inducing inflammatory responses, disrupting barrier function, and triggering oxidative stress [3]. Studies have reported that LPS induced inflammation in goslings can result in severe intestinal injury and liver damage, impairing the overall health and production performance of goslings. Furthermore, LPS-induced intestinal inflammation is associated with liver damage, which exacerbates the deterioration of systemic health. Prior studies have demonstrated that LPS-induced intestinal damage is linked to gut microbiota dysregulation, which aggravates inflammation and disrupts gut functionality [4]. Apart from its harmful effects on intestinal health, LPS accumulation has been associated with various diseases, including diarrheal diseases, in different species. LPS is a key factor in triggering diarrhea by promoting immune activation and intestinal fluid secretion. Research indicates that elevated LPS levels can lead to gut inflammation and alter gut microbiota composition, ultimately contributing to diarrhea and other gastrointestinal disorders. For example, LPS-induced accumulation can impair gut integrity and trigger symptoms such as diarrhea in both animals and humans [5]. These findings underscore the importance of LPS in the pathogenesis of intestinal dysfunction, providing a basis for exploring therapeutic strategies to alleviate LPS-induced gut injury.

The intestinal microecosystem comprises the gut microbiota, epithelial cells, and the mucosal immune system to form a complex and interactive network. The gut microbiota plays a crucial role in maintaining intestinal homeostasis, regulating host metabolism and immune functions and contributing to disease pathogenesis [6]. Studies have reported that the intestinal barrier prevents the translocation of LPS into the systemic circulation; however, increased LPS permeability exacerbates local inflammatory responses during intestinal inflammatory diseases such as inflammatory bowel disease (IBD) [7]. Furthermore, gut microbiota dysbiosis may adversely affect the immune system and increase the risk of autoimmune diseases such as rheumatoid arthritis [8]. Disturbances in the intestinal microecological balance can lead to various conditions such as obesity, diabetes, and IBD [9]. Furthermore, gut microbiota dysbiosis can contribute to disease development through multiple mechanisms. Therefore, maintaining gut microbial homeostasis is crucial.

Metabolomics technology has prominently emerged in biomedical research owing to its ability to comprehensively analyze small-molecule metabolites in biological systems, directly reflecting the functional output of microbial communities and their regulatory effects on metabolism [10]. The gut is not only the primary site for nutrient digestion and absorption but also a key metabolic organ where various biochemical reactions occur. Changes in gut metabolism are closely linked to its digestive and absorptive functions, and metabolomics provides an effective means to assess alterations in metabolic pathways, which, in turn, reflects functional changes in the gut microenvironment.

Untargeted metabolomics has been used to successfully identify increased levels of deoxycholic acid and palmitamide in the serum of patients with Crohn's disease and IBD, highlighting the crucial role of bile acid synthesis and fatty acid metabolism disorders in disease progression [11]. This finding underscores the sensitivity of metabolomics in identifying disease

specific metabolic markers and pathway perturbations. Similarly, in an antibiotic-induced gut microbiota-perturbation model, the integration of 16S rRNA sequencing with metabolomics revealed that the fermentation products of pineapple whey could restore the function of gut microbiota and alleviate intestinal damage by upregulating amino acid metabolic pathways including the L-glutamate and L-threonine pathways. These examples emphasize the necessity of multiomics integration in deciphering complex biological processes [12]. In this study, metabolomics was used to determine the metabolic changes induced by PAMK supplementation, as it is administered orally and primarily exerts its effects via the digestive tract before its systemic absorption. As PAMK is metabolized in the gut, its biological effects are reflected in the changes in metabolite profiles. By integrating metabolomics with 16S rRNA sequencing, it is possible to explore how PAMK regulates microbial metabolism, influences host metabolic pathways, and ultimately improves gut function.

In Shennong Bencao Jing (The Classic of Materia Medica), *Atractylodes macrocephala* Koidz (AMK) is described as a medicinal plant with various therapeutic effects, including strengthening the spleen and stomach, eliminating edema, and enhancing immunity. PAMK is an active ingredient derived from AMK and is known for its immunomodulatory, gut environment-improving, anti inflammatory, and antioxidant effects [13]. Studies have reported that PAMK significantly ameliorates disruptions in the composition of the gut microbiota, enhances microbial diversity indices, and restores the structure of the gut microbiota [14]. Moreover, studies have reported that PAMK enhances intestinal mucosal immune function by increasing the total levels of secretory immunoglobulin A, thereby improving the immune response of the body [15]. Previous studies have demonstrated that PAMK alleviates LPS-induced gut dysbiosis by increasing the abundance of beneficial bacteria, such as Romboutsia, thereby exerting protective and restorative effects on the intestine [16]. These findings align with the traditional uses of PAMK in traditional Chinese medicine (TCM), where it is used to improve gut health and immune function. In this study, the protective effects of PAMK in LPS-induced intestinal injury in goslings were investigated using in vivo assays. This approach allowed us to evaluate the efficacy of PAMK in a physiological context, providing insights into its potential therapeutic applications in maintaining the intestinal health of poultry and preventing intestinal diseases.

Previous studies by our research group have investigated the effects of different PAMK concentrations on the gut microbiota and immune function in goslings, and 400 mg/kg PAMK was found to be effective in modulating the gut microbiota and enhancing immune responses without causing any adverse effects. Similarly, an LPS concentration of 2 mg/kg body weight was found to induce significant intestinal injury in goslings [17,18]. These concentrations were therefore chosen for the current study to evaluate the protective effects of PAMK against LPS-induced intestinal injury.

The potential mechanisms underlying the protective effects of PAMK were elucidated in this study by integrating 16S rRNA analysis and metabolomics. This study provides scientific evidence highlighting the role of PAMK as a therapeutic agent in mitigating LPS-induced intestinal injury and offers new strategies for maintaining the intestinal health of poultry and preventing intestinal diseases.

## 2 Materials and methods

### 2.1 Animals grouping and treatments

One-day-old Magang goslings (Anser cygnoides) (*n* = 80, half males and half females) were purchased from Guangdong Qing yuan Jin yu feng Goose Co., Ltd. Magang goslings were randomly divided into four groups (*n* = 20): the control group, PAMK group, LPS group

and P-LPS group. The control and LPS groups were fed a standard diet, while the diets of the PAMK and P-LPS groups were supplemented with 400 mg/kg PAMK. During the rearing period, the goslings had ad libitum access to feed and water. The animals were monitored daily for any signs of distress or illness, and appropriate veterinary care was provided as needed. As shown in Table 1, on days 24, 26, and 28 of age, the LPS and P-LPS groups were intraperitoneally injected with LPS solution at a dose of 2 mg/kg·BW, while the other two groups received 0.5 mL of physiological saline. Ten 28-day-old goslings were randomly selected from each group, the goslings were euthanized by intraperitoneal injection of a lethal dose of sodium pentobarbital (100 mg/kg·BW) under deep anesthesia to minimize pain and distress. The animals were confirmed dead by cessation of heart activity and lack of respiration following the injection. All euthanasia procedures were carried out under the supervision of trained personnel to ensure humane treatment. Serum, cecal contents, and jejunum samples were collected from the goslings. All samples were immediately placed in liquid nitrogen and stored at –80°C for further analysis. The experiment received prior ethical approval in accordance with Zhongkai University of Agriculture and Engineering and under the approved protocol number ZK202110-10, with animal qualification number NO20210906.

## 2.2 Reagents

PAMK (purity 95%, product code: CY201216) was purchased from Yangling Ciyuan Biotechnology Co., Ltd. in Shaanxi Province, China; LPS (product code: L2880-50MG) was obtained from Sigma, located in St. Louis, Missouri, USA; Trizol RNA isolation reagent (product code: GK20008) was sourced from GLPBIO, located in Montclair, California, USA; reverse transcription reagent (product code: A304-10) was purchased from Beijing Kangrun Biology Company in Beijing, China; PowerUp™ SYBR™ Green Master Mix (product code: A25742) was purchased from Applied Biosystems, located in Waltham, Massachusetts, the United States; IL-1$\beta$(product code: ZK-G7321), IL-6(product code: ZK-G7318), IL-18(product code: ZK-G7323), PCT(product code: ZK-G7512), CRP(product code: ZK-G7610), TNF-$\alpha$(product code: ZK-G7319)ELISA kits were purchased from Shenzhen Zike Biotechnology Company in Shenzhen, China.

## 2.3 Detection of serum inflammatory cytokines

The levels of inflammatory cytokines, including IL-1$\beta$, IL-6, IL-18, TNF-$\alpha$, PCT, and CRP in gosling serum were measured according to the instructions provided with the ELISA kits.

## 2.4 16S rRNA sequencing

Ten 28-day-old goslings per group were randomly selected for sample collection. After euthanizing the goslings with a lethal d ose of sodium pentobarbital (100 mg/kg·BW) under

**Table 1. Experimental design for dietary and injection treatments.**

| Group | Number ($n$) | Dietary Treatment | Injection Treatment |
|---|---|---|---|
| Control | 20 | Standard diet | i.p.[1] 0.5 mL NS[2] |
| PAMK | 20 | Standard diet + 400 mg/kg PAMK | i.p. 0.5 mL NS |
| LPS | 20 | Standard diet | i.p. 2 mg/kg LPS |
| P-LPS | 20 | Standard diet + 400 mg/kg PAMK | i.p. 2 mg/kg LPS |

**Note:** [1] i.p, intraperitoneal injection.2 NS, physiological saline.

deep anesthesia, the abdominal area was disinfected, and the cecal contents were collected aseptically. Sterile surgical tools were used to carefully isolate the cecum, and contents were transferred into sterile cryotubes (1.5 mL per tube). Samples with abnormal characteristics (e.g., unusual color or odor) were discarded. were discarded, and the tubes were immediately placed in liquid nitrogen for rapid freezing. Six randomly selected samples from each group were used for 16S rRNA sequencing.

Total genomic DNA samples were extracted using the OMEGA Soil DNA Kit (M5635-02) (Omega Bio-Tek, Norcross, GA, USA), following the manufacturer's instructions, and stored at −20°C prior to further analysis. The quantity and quality of extracted DNAs were measured using a NanoDrop NC2000 spectrophotometer (Thermo Fisher Scientific, Waltham, MA, USA) and agarose gel electrophoresis, respectively. The V3-V4 region of the bacterial 16S rRNA gene was PCR-amplified using the forward primer 338F (5'-ACTCCTACGGGAGGCAGCA-3') and the reverse primer 806R (5'-GGACTACHVGGGT WTCTAAT-3'). To enable multiplex sequencing, a sample-specific 7-base barcode was added to the primers. The PCR amplicons were purified using Vazyme VAHTSTM DNA Clean Beads and quantified using the Quant-iT PicoGreen dsDNA Assay Kit. After the quantification step for individual samples, the amplicons were mixed in equal volumes and subjected to $2 \times 250$ bp paired-end sequencing using the Illumina TruSeq Nano DNA LT Library Preparation Kit at Suzhou Panomics Biotechnology Co., Ltd. in Suzhou, China. The raw high-throughput sequencing data were initially screened based on quality. Problematic samples were re-sequenced or additional sequencing was performed. The qualified sequences after initial screening were assigned to libraries and samples according to the index and barcode, and the barcode sequences were removed. The QIIME2 dada2 or Vsearch pipeline was used for sequence denoising or OTU clustering. The species composition of each sample (group) at different taxonomic levels was presented to understand the general situation. The Alpha diversity of the samples was evaluated based on the ASV/OTU distribution, and rarefaction curves were used to assess the sequencing depth. At the ASV/OTU level, the sample distance matrix was calculated, and ordination, clustering, and statistical tests were used to measure the differences and significance of beta diversity between samples (groups). At the species taxonomic composition level, various analysis methods and statistical tests were used to measure the differences in species abundance between samples (groups) and to identify marker species. An association network was constructed based on the species distribution, topological indices were calculated, and key species were identified. Based on the sequencing results of 16S rRNA, 18S rRNA, and ITS, the metabolic functions of the microbial community were predicted, and the differential pathways and the species composition of specific pathways were identified.

## 2.5 Untargeted metabolomics sequencing

**2.5.1 Sample preparation.** Thaw the experimental sample at a 4°C, vortex the sample for 1 min after thawing, and mix evenly. Accurately transfer an appropriate amount of sample into a 2 mL centrifuge tube. Add 400 μL methanol (stored at −20°C) and vortex for 1 min. Centrifuge for 10 min at 12,000 rpm and 4°C, take all the supernatant transfer it to a new 2 mL centrifuge tube, concentrate and dry it. Add 150 μL of 2-chloro-l-phenylalanine (4 ppm) solution prepared with 80% methanol water (stored at 4°C) to redissolve the sample, filter the supernatant by 0.22 μm membrane and transfer into the detection bottle for LC-MS detection.

**2.5.2 Liquid chromatography conditions.** The Thermo Vanquish ultra-high-performance liquid chromatography (UHPLC) system (Thermo Fisher Scientific, USA) was used for chromatographic separation. An ACQUITY UPLC ® HSS T3 column (2.1 × 150 mm,

1.8 μm; Waters, Milford, MA, USA) was employed with a flow rate of 0.25 mL/min at a column temperature of 40°C. The injection volume was set to 2 μL. For positive ion mode, the mobile phases consisted of 0.1% formic acid in acetonitrile (C) and 0.1% formic acid in water (D). The gradient elution program was as follows: 0–1 min, 2% C; 1–9 min, 2%–50% C; 9–12 min, 50%–98% C; 12–13.5 min, 98% C; 13.5–14 min, 98%–2% C; 14–20 min, 2% C. For negative ion mode, the mobile phases consisted of acetonitrile (A) and 5 mM ammonium formate in water (B). The gradient elution program was as follows: 0–1 min, 2% A; 1–9 min, 2%–50% A; 9–12 min, 50%–98% A; 12–13.5 min, 98% A; 13.5–14 min, 98%–2% A; 14–17 min, 2% A.

**2.5.3 Mass spectrum conditions.** The Thermo Orbitrap Exploris 120 mass spectrometer (Thermo Fisher Scientific, USA) equipped with an electrospray ionization (ESI) source was used for data acquisition in both positive and negative ion modes. The spray voltage was set to 3.50 kV for positive ion mode and –2.50 kV for negative ion mode. The sheath gas flow rate was 30 arb, and the auxiliary gas flow rate was 10 arb. The capillary temperature was maintained at 325°C. Full MS scans were acquired at a resolution of 60,000 over an m/z range of 100-1000. Higher-energy collisional dissociation (HCD) was employed for tandem MS (MS/MS) fragmentation with a collision energy of 30%. The MS/MS resolution was set to 15,000, and the top four most intense precursor ions were selected for fragmentation. Dynamic exclusion was applied to remove unnecessary MS/MS information.

## 2.6 Real-time quantitative PCR

Total RNA was extracted from jejunal tissues of goslings using the Trizol RNA extraction kit, followed by reverse transcription into cDNA. Primers were designed based on gene sequences of geese or closely related species available on the NCBI website and synthesized by Sangon Biotech (Shanghai, China). The gene-specific primer sequences used in this study are listed in Table 2. Quantitative real-time PCR (qRT-PCR) was performed using the QuantStudio 7 Real-Time PCR System. The relative mRNA expression levels of target genes were calculated using the $2^{-\Delta\Delta ct}$ method.

## 2.7 Statistical analysis

All data were analyzed using SPSS statistical software (Version 20.0 for Windows, SPSS, Chicago, IL) through one-way analysis of variance (ANOVA). Results are presented as mean ± standard deviation (SD). Data visualization was performed using GraphPad Prism 9.5.1. A value of $P < 0.05$ was considered to indicate statistical significance.

**Table 2. Primer sequence.**

| Gene | GenBank Accession | Forward Primer(5'-3') | Reverse Primer(5'-3') |
|---|---|---|---|
| $\beta$-actin | XM_066977989.1 | GCACCCAGCACGATGAAAAT | GACAATGGAGGGTCCGGATT |
| IL-1$\beta$ | XM_048054160.2 | TCATCTTCTACCGCCTGGAC | GTAGGTGGCGATGTTGACCT |
| IL-6 | XM_048070285.2 | GATCCGGCAGATGGTGATAA | AGGATGAGGTGCATGGTGAT |
| IL-18 | XM_048063324.2 | TGCCTCTACTTTGCTGACGA | ACCACAAGCACCTGGCTATT |
| ZO-1 | XM_048060267.2 | CGACTCCTCGTCGGGTGA | ACTGAGACACAGTTTGCTCCA |
| TLR-2 | XM_048071758.2 | CCACGCAGTCTGGTACATGA | CACCCAGTTGGAGTCGTTCT |
| Claudin | XM_013199194.3 | GACCAGGTGAAGAAGATGCGGATG | CGAGCCACTCTGTTGCCATACC |
| Mucin-2 | XM_066997591.1 | GTCAGTCATGGTGGCCGTGTAAC | CGTCATCAAGGACTTGCACAGGAG |
| Occludin | XM_066983782.1 | CAGGATGTGGCAGAGGAATACAA | CCTTGTCGTAGTCGCTCACCAT |

## 3 Results

### 3.1 The effects of PAMK on serum inflammatory cytokine secretion in LPS-treated goslings

IL-1$\beta$, IL-6, IL-18, and tumor necrosis factor (TNF)-$\alpha$ are pro inflammatory cytokines, and procalcitonin (PCT) and C-reactive protein (CRP) are nonspecific markers of systemic inflammation that increase sharply during pathogenic infections including bacterial infections. The relative levels of IL-1$\beta$, IL-6, IL-18, TNF-$\alpha$, PCT, and CRP in gosling serum were determined. The LPS group exhibited significantly higher IL-1$\beta$, IL-6, TNF-$\alpha$, and CRP levels ($P < 0.05$) compared with the control group, indicating that LPS induced an inflammatory response in the intestines of goslings (Fig 1). Compared with the LPS group, the P-LPS group showed significantly lower levels of IL-6 and CRP ($P < 0.05$), whereas IL-1$\beta$, IL-18, TNF-$\alpha$, and PCT levels exhibited a downward trend, which were not significant ($P < 0.05$). Furthermore, compared with the control group, the PAMK group exhibited significantly lower levels

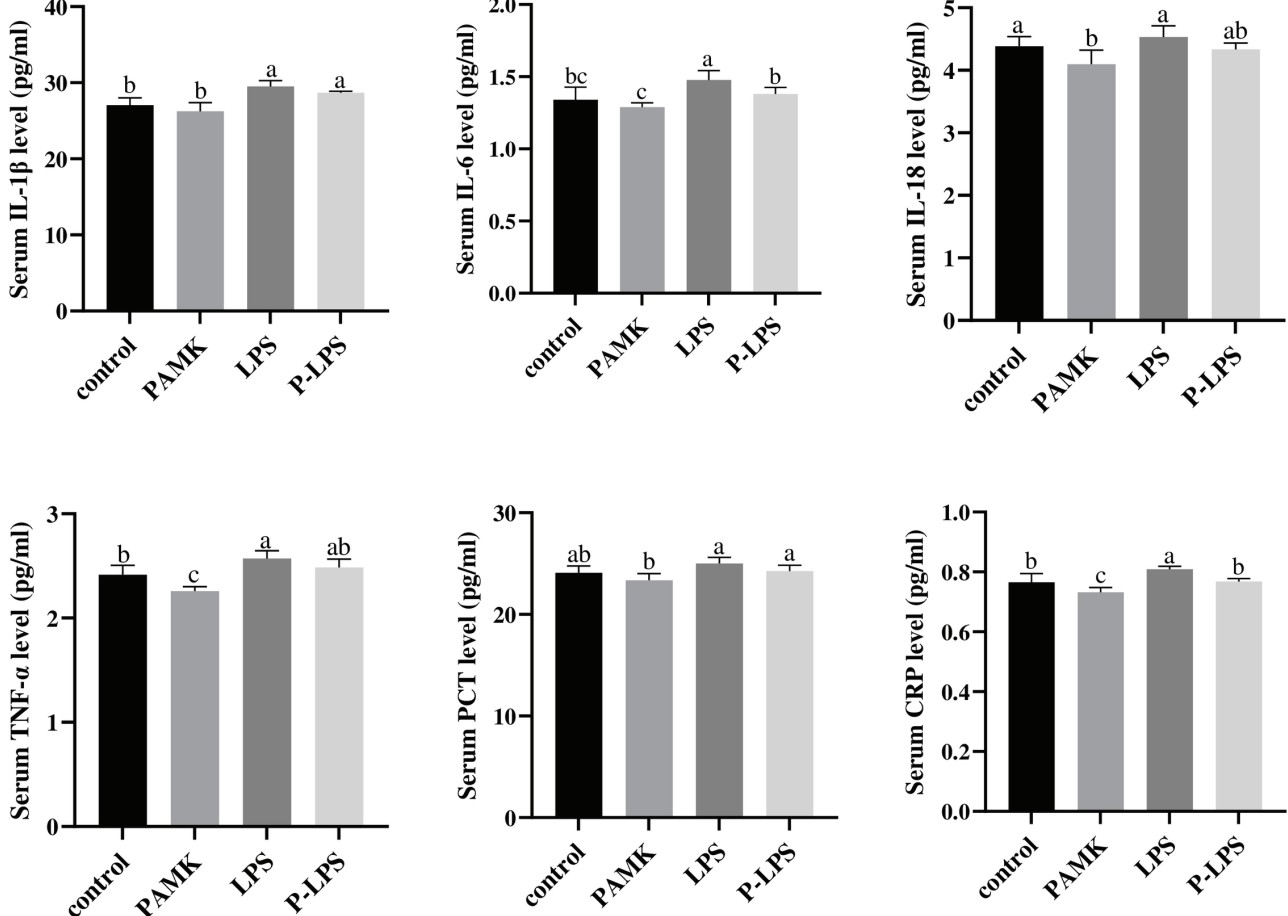

**Fig 1. Serum ELISA detection Inflammatory cytokines IL-1$\beta$, IL-6, IL-18, and inflammatory markers TNF-$\alpha$, PCT, and CRP ($n$ = 9).** Data are shown with mean ±SD. Different letters above the bars indicate significant differences ($P < 0.05$), while no letters or the same letters indicate no significant difference ($P > 0.05$). The same applies to the table below.

of IL-18, TNF-$\alpha$, and CRP ($P < 0.05$), whereas IL-1$\beta$, IL-6, and PCT levels were not significantly different ($P < 0.05$). These findings indicate that LPS induces systemic inflammation in goslings, which can be mitigated by PAMK.

## 3.2 PAMK exhibits protective effects against LPS-induced intestinal injury

mRNA expression of inflammatory cytokines and the LPS receptor Toll-like receptor (TLR)-2 in the jejunal tissues of goslings was determined. Compared with the control group, the LPS group exhibited a significant increase in the relative mRNA expression of *IL-1$\beta$* and *TLR-2* ($P < 0.05$), suggesting the LPS-induced inflammatory response in the jejunum of goslings (Fig 2). Compared with the LPS group, the P-LPS group exhibited a significantly lower mRNA expression of *IL-1$\beta$*, *IL*-6, and *TLR*-2 ($P < 0.05$), with a downward trend observed for *IL*-18 ($P < 0.05$). These findings indicate that LPS triggered an inflammatory response in the jejunal tissues of goslings, which could be alleviated by PAMK.

Intestinal inflammation can compromise the integrity of the intestinal barrier. Therefore, the mRNA expression of tight junction proteins *ZO-1*, *Occludin*, *Claudin*, and *Mucin-2* in the jejunum was assessed. The LPS group exhibited a significant decrease in the relative mRNA

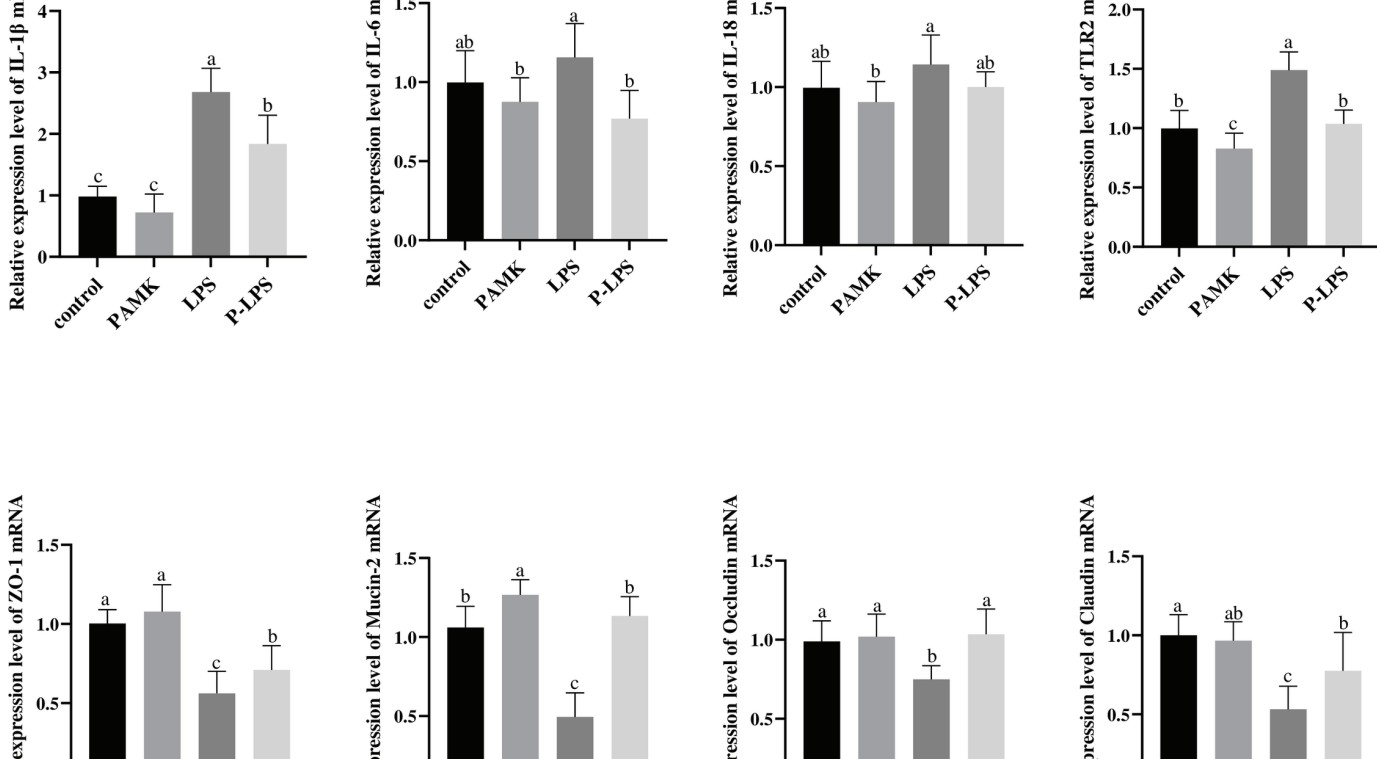

**Fig 2. Jejunal gene expression.** Relative mRNA expression levels of inflammatory cytokines and tight junction proteins ($n = 7$). Data are shown with mean ± SD.

expression of *ZO*-1, *Occludin*, *Claudin*, and *Mucin*-2 (*P* < 0.05) compared with the control group, indicating that LPS disrupted intestinal barrier function (Fig 2). However, the P-LPS group showed a significant increase in the mRNA expression of *ZO*-1, *Occludin*, *Claudin*, and *Mucin*-2 (*P* < 0.05) compared with the LPS group. These findings demonstrate that PAMK could effectively alleviate LPS-induced intestinal barrier damage in goslings.

## 3.3 PAMK alleviates LPS-induced intestinal microbiota dysbiosis

**3.3.1 PAMK modulates the overall structure of intestinal microbiota in LPS-treated goslings.** 16S rRNA high-throughput sequencing was used to determine whether PAMK could alleviate LPS-induced changes in the gut microbiota of goslings. Principal component analysis (PCA; Fig 3A) revealed the control, PAMK, and P-LPS groups to be tightly clustered and the LPS group to be distant from these groups, indicating the microbiota structure in the control, PAMK, and P-LPS groups to be similar and suitable for further analysis. The Chao1 index reflects species richness as shown in Fig 3B. The Chao1 index of the PAMK group did not change significantly compared with that of the control group. The LPS group showed a downward trend in the Chao1 index, but the difference was not significant. The P-LPS group exhibited a significant increase in the Chao1 index (*P* < 0.05) compared with that of the LPS group. The Simpson and Shannon indices reflect species diversity. The Simpson and Shannon indices of the LPS group tended to decrease compared with those for the control group, but the differences were not significant (*P* < 0.05). The P-LPS group showed an upward trend in the Simpson and Shannon indices compared with those in the LPS group, but no significant differences were noted (*P* < 0.05). These results suggest that PAMK could alleviate the LPS-induced reduction in gut microbiota abundance and the decrease in microbial diversity in goslings.

Next, the relative abundance of gut microbiota in goslings was analyzed at the phylum level. The Venn diagram (Fig 3C) shows 1194 overlapping Operational Taxonomic Units (OTUs) across the four groups. The control and PAMK groups share 2718 OTUs, the control and LPS groups share 3094 OTUs, the control and P-LPS groups share 2543 OTUs, and the LPS and P-LPS groups share 2198 OTUs. Firmicutes and Bacteroidetes were the most abundant across all samples at the phylum level (Fig 3D). The LPS group exhibited an increase in Bacteroidetes and a decrease in Proteobacteria abundance compared with that in the control group. In contrast, the P-LPS group demonstrated a decrease in the abundance of Firmicutes and an increase in the abundance of Bacteroidetes compared with that in the LPS group. These findings indicate that PAMK could alleviate the LPS-induced changes in gut microbiota abundance. The LPS group showed a significant increase in *Bacteroides* and *Lactobacillus* at the genus level (Fig 3E), whereas the P-LPS group showed no significant changes compared with that in the control group. The LPS group also exhibited a decrease in the abundance of *Barnesiella*, *Faecalibacterium*, and *Oscillospira* compared with that in the control group. In contrast, the P-LPS group demonstrated an increased abundance of *Faecalibacterium* and *Oscillospira*, suggesting the beneficial modulation of the gut microbiota by PAMK.

**3.3.2 PAMK modulates differential gut microbial taxa in response to LPS-induced intestinal injury.** Linear Discriminant Analysis Effect Size analysis was conducted to determine the impact of PAMK on gut microbiota composition, and distinct bacterial profiles were noted among the groups (Fig 4). In the PAMK group, the dominant taxa included *g_Faecalibacterium*, *f_Desulfovibrionaceae*, and *g_Oxalobacter*. Conversely, the LPS group exhibited significant enrichment of *g_Melissococcus*, *f_Enterococcaceae*, and *f_Xanthobacteraceae*. In the P-LPS group, *f_Rikenellaceae*, *g_Oscillospira*, and *g_Megamonas* were the predominant bacteria. *g_Faecalibacterium* is commonly associated with a healthy gut

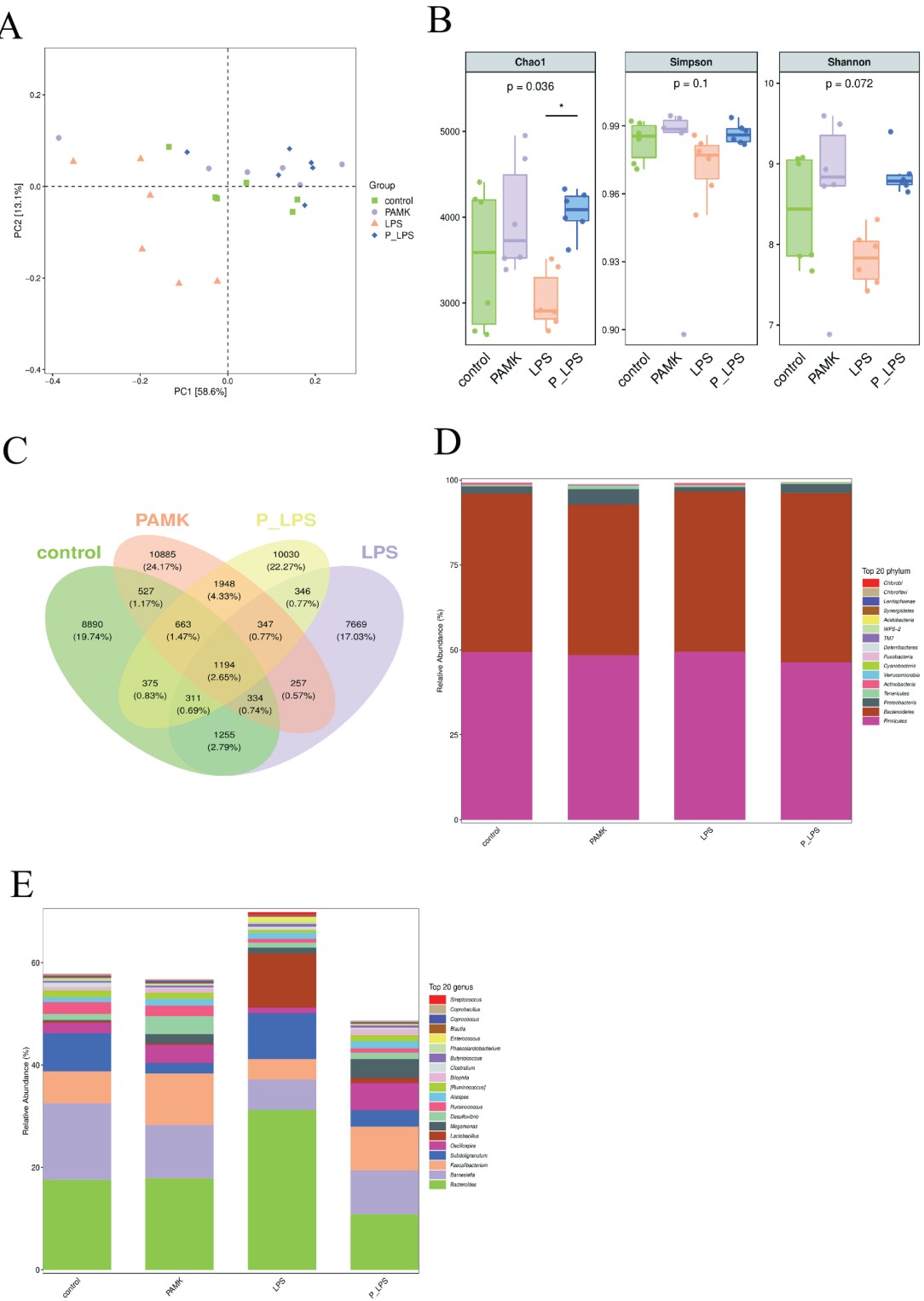

**Fig 3. Microbial community diversity analysis** (*n* = 6)**.** (A) The PCA analysis two-dimensional ordination plot of the samples. Each point in the figure represents a sample, with different colors indicating different groups. (B) Gut microbiota alpha diversity indices.Each panel corresponds to an alpha diversity index, indicated in the gray area at the top. (C) Venn diagram of ASV/OTU across groups. The numbers in each section indicate the number of ASVs/OTUs contained in that section. (D) Bar chart of species composition at the phylum level. (E) Bar chart of species composition at the genus level.

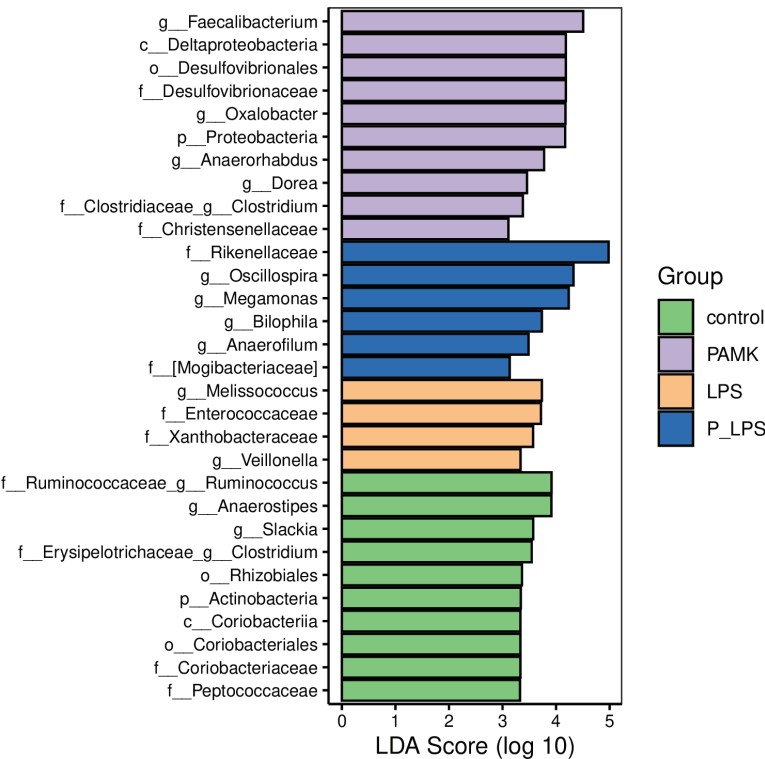

**Fig 4. Biomarker species of each group (*n* = 6).** The bar plot shows the distribution of LDA scores for significantly different taxa. The y-axis represents the taxa with significant differences between groups, while the x-axis visually displays the log-transformed LDA scores for each taxonomic unit. The taxa are ranked by their LDA scores, indicating their specificity in sample grouping. The longer the bar, the more significant the difference for that taxon. The color of the bars corresponds to the group in which the respective taxonomic unit exhibits the highest abundance.

microbiota, contributing to gut barrier integrity and anti-inflammatory responses. Its enrichment in the PAMK group suggested that PAMK may promote the growth of healthier gut microbiota. Additionally, *g_Megamonas*, a genus within the Bacteroidetes phylum, is linked to gut health, further indicating that PAMK supplementation may support the regulation and restoration of the gut microbial balance.

### 3.4 Non-targeted metabolomics of gosling serum

**3.4.1 Multivariate analysis confirms distinct metabolic profiles between LPS and P-LPS groups.** Quality control (QC) samples were used for liquid chromatography-mass spectrometry (LC-MS) to ensure data reliability. Theoretically, QC samples are identical, but systematic errors during sample extraction and analysis may introduce variations. Smaller variations indicate higher method stability and better data quality. This is reflected by the tight clustering of QC samples in the PCA plot, demonstrating data reliability. PCA was performed using the metabolite data from the LPS and P-LPS groups of goslings (Fig 5A). Intestinal tissue samples from the LPS and P-LPS groups were well separated, and the samples in the same group were clustered closely together. As shown in Fig 5B, a supervised partial least squares discriminant analysis (PLS-DA) model was used to evaluate the differences between groups. The PLS-DA score plots show that the P-LPS group vs LPS group had $R^2Y = 0.997$ and $Q^2 = 0.666$ in the positive ion mode and $R^2Y = 0.999$ and $Q^2 = 0.601$ in the negative ion mode. QC

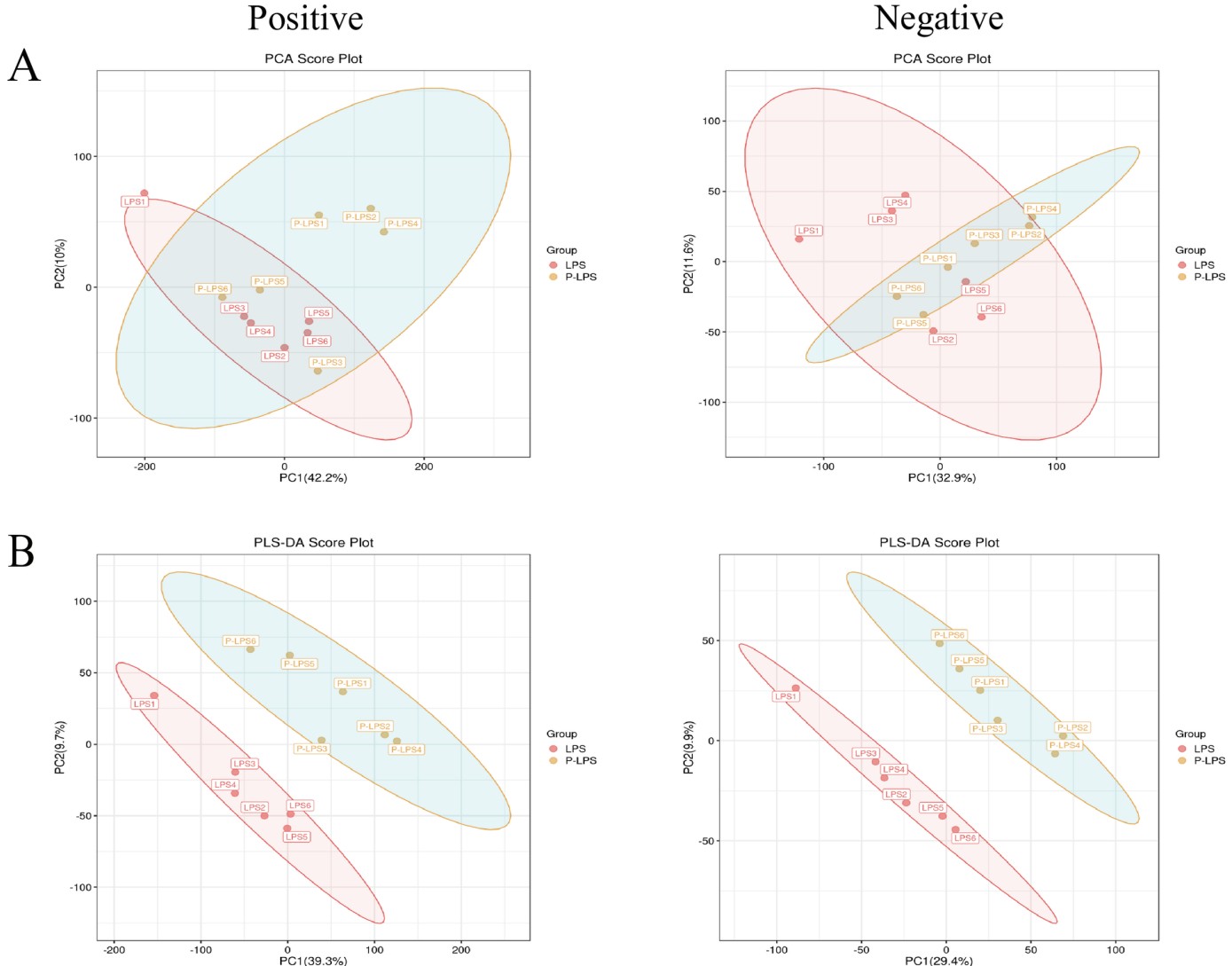

**Fig 5. PCA (A) and PLS-DA (B) analyses were performed on serum samples from the P-LPS group vs the LPS group in both positive and negative ion modes (*n* = 6).** The x-axis represents the explained variance of the first principal component, and the y-axis represents the explained variance of the second principal component. Points represent experimental samples, with colors indicating different groups. The more clustered the samples within a group and the more dispersed the samples between groups, the more reliable the results.

samples were used to ensure data reliability for LC-MS analysis. PCA showed a clear separation between the LPS and P-LPS groups, whereas PLS-DA confirmed significant metabolic differences and high model reliability.

### 3.4.2 Identification of differential metabolites between P-LPS and LPS groups.

First, metabolite identification was confirmed based on accurate molecular mass, followed by annotation using MS/MS fragmentation patterns. This was cross-referenced with databases including the Human Metabolome Database (http://www.hmdb.ca), MassBank (http://www.massbank.jp), LipidMaps (http://www.lipidmaps.org), mzCloud (https://www.mzcloud.org), and an in-house standard metabolite database. Differential

metabolites in the serum were identified from the primary metabolite list by applying pre-set thresholds for the *p*-value and variable importance in projection score during statistical analysis. As shown in Table 3 , a total of 373 differential metabolites were identified, among which 24 were significantly altered in the P-LPS group compared with those in the LPS group (Table 4). Of these, 3 metabolites were upregulated and 21 were downregulated. These findings highlight the role of PAMK in regulating metabolic pathways, particularly in alleviating LPS-induced intestinal injury by significantly altering serum metabolites, with the majority being downregulated.

**3.4.3 Mass spectra of key marker substances in the comparison between P-LPS and LPS groups.** The samples of the P-LPS vs LPS group were analyzed using LC-MS/MS (Fig 6), and the mass spectra and identification information of key marker substances including L-aspartic acid, S-adenosylmethionine, itaconic acid, and L-octanoylcarnitine were obtained (Table 5).

**Table 3. Statistical table of differential metabolites.**

| Comparison | Total | Up | Down | Total_DE |
|---|---|---|---|---|
| PAMK vs control | 373 | 5 | 10 | 15 |
| LPS vs control | 373 | 16 | 13 | 29 |
| P-LPS vs control | 373 | 11 | 24 | 35 |
| P-LPS vs LPS | 373 | 3 | 21 | 24 |
| P-LPS vs PAMK | 373 | 17 | 33 | 50 |
| LPS vs PAMK | 373 | 44 | 13 | 57 |

**Note:** 'Total' indicates Total number of metabolites. 'Total_DE' represents the differential metabolites between the two groups. 'Up' indicates upregulated metabolites, and 'Down' indicates downregulated metabolites.

**Table 4. The metabolic differences between P-LPS and LPS.**

| Name | Formula | KEGG | FDR | VIP | Pos/Neg |
|---|---|---|---|---|---|
| Epsilon-caprolactam | C6H11NO | C06593 | 0.61583 | 2.369083 | Pos |
| Dihydrouracil | C4H6N2O2 | C00429 | 0.620707 | 1.680032 | Pos |
| 1,2,3-Trihydroxybenzene | C6H6O3 | C01108 | 0.620707 | 2.184746 | Pos |
| 5,6-Dihydro-5-fluorouracil | C4H5FN2O2 | C16630 | 0.620707 | 1.479624 | Pos |
| 6-Hydroxyhexanoic acid | C6H12O3 | C06103 | 0.620707 | 1.276708 | Pos |
| L-Aspartic acid | C4H7NO4 | C00049 | 0.620707 | 1.259833 | Pos |
| 12-Hydroxydodecanoic acid | C12H24O3 | C08317 | 0.620707 | 1.72623 | Pos |
| Pyrimidodiazepine | C9H11N5O2 | C02587 | 0.620707 | 1.601448 | Pos |
| L-Octanoylcarnitine | C15H29NO4 | C02838 | 0.620707 | 2.063902 | Pos |
| Dehypoxanthine futalosine | C14H16O7 | C17010 | 0.620707 | 1.896912 | Pos |
| Erucic acid | C22H42O2 | C08316 | 0.620707 | 1.443849 | Pos |
| Erucic acid | C22H42O2 | C08316 | 0.620707 | 1.443849 | Pos |
| Sucrose | C12H22O11 | C00089 | 0.620707 | 2.060385 | Pos |
| AMP | C10H14N5O7P | C00020 | 0.620707 | 1.582982 | Pos |
| Nitrendipine | C18H20N2O6 | C07713 | 0.620707 | 1.749685 | Pos |
| S-Adenosylmethionine | C15H22N6O5S | C00019 | 0.620707 | 1.954449 | Pos |
| N2'-Acetylgentamicin C1a | C21H41N5O8 | C03524 | 0.620707 | 1.300779 | Pos |
| 3-Methylthiopropionic acid | C4H8O2S | C08276 | 0.716359 | 1.860387 | Neg |
| Itaconic acid | C5H6O4 | C00490 | 0.683322 | 2.07692 | Neg |
| Beta-Guanidinopropionic acid | C4H9N3O2 | C03065 | 0.716359 | 1.747553 | Neg |
| Glycyl-glycine | C4H8N2O3 | C02037 | 0.716359 | 1.886549 | Neg |
| beta-D-Fructose | C6H12O6 | C02336 | 0.716359 | 1.929754 | Neg |
| 3-Methylxanthine | C6H6N4O2 | C16357 | 0.683322 | 2.106121 | Neg |
| Capric acid | C10H20O2 | C01571 | 0.716359 | 1.679207 | Neg |
| Anserine | C10H16N4O3 | C01262 | 0.716359 | 2.138263 | Neg |

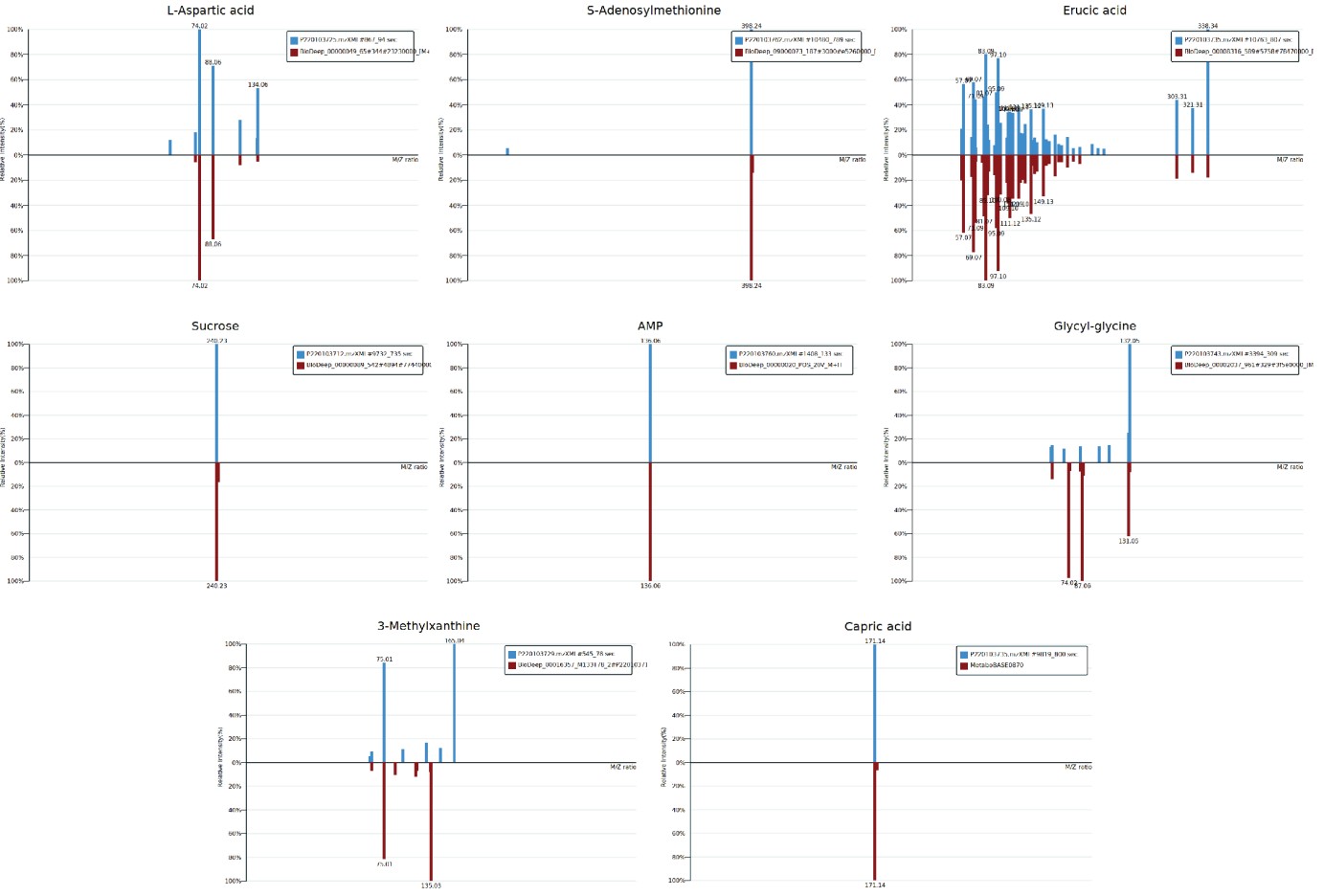

**Fig 6. Mass spectra of key metabolites in the comparison between P-LPS and LPS groups** (*n* = 6)**.** The blue section represents the actual spectrum of the substance, while the red section represents the theoretical spectrum from the database. The X-axis represents the mass-to-charge ratio (m/z) of the molecular ions, reflecting the ion characteristics of different molecules. The Y-axis represents the ion intensity at that m/z value, which typically reflects the relative abundance or concentration of the molecules.

**Table 5. Information on mass spectra of key marker substances.**

| Name | Formula | KEGG | FC | FDR | VIP | Pos/Neg |
|---|---|---|---|---|---|---|
| L-Aspartic acid | C4H6N2O | C00429 | 0.71 | 0.620707 | 1.259833 | Pos |
| S-Adenosylmethionine | C15H22N6O5S | C00019 | 0.65 | 0.620707 | 1.954449 | Pos |
| Erucic acid | C22H42O2 | C08316 | 0.48 | 0.620707 | 1.443849 | Pos |
| Sucrose | C12H22O11 | C00089 | 0.48 | 0.620707 | 2.060385 | Pos |
| AMP | C10H14N5O7P | C00020 | 0.72 | 0.620707 | 1.582982 | Pos |
| Itaconic acid | C5H6O4 | C00490 | 0.67 | 0.683322 | 2.07692 | Neg |
| Glycyl-glycine | C4H8N2O3 | C02037 | 0.64 | 0.716359 | 1.886549 | Neg |
| 3-Methylxanthine | C6H6N4O2 | C16357 | 0.19 | 0.683322 | 2.106121 | Neg |
| Capric acid | C10H20O2 | C01571 | 0.62 | 0.716359 | 1.679207 | Neg |

Comparisons of MS data of the samples with the known standard substances were consistent, confirming the presence of the target compounds in the samples.

**3.4.4 Differential metabolic pathways indicating PAMK's role in alleviating LPS-induced toxicity.** The differential metabolic pathways between sample groups were analyzed

to elucidate the role of PAMK in alleviating the toxic effects of LPS by regulating metabolism. MetPA, a component of MetaboAnalyst, is primarily based on the Kyoto Encyclopedia of Genes and Genomes (KEGG) metabolic pathways. Based on pathway enrichment and topology analysis, the MetPA database can identify metabolic pathways that are potentially affected by biological perturbations, enabling the analysis of associated metabolites. As shown in Fig 7A, 11 pathways with significant differences ($P < 0.05$) were identified between the LPS and control groups. These pathways included serotonin receptor agonist/antagonist pathway, cyclic adenosine monophosphate signaling pathway, and linoleic acid metabolism. In terms of the inflammatory response and intestinal barrier function, serotonin in the serotonin receptor agonist/antagonist pathway is involved in regulating intestinal physiological functions. PAMK may alleviate inflammation and protect the intestinal barrier by regulating this pathway. The metabolites produced during the linoleic acid metabolism pathway can affect gut microbiota balance. LPS disrupts this metabolic process, whereas PAMK may improve the microbial ecological environment by regulating this pathway. The $\beta$-alanine metabolism pathway is related to energy metabolism and the synthesis of bioactive substances. LPS interferes with its metabolism, and PAMK may provide energy to intestinal cells and promote damage repair by regulating this pathway. The differential metabolic pathways between the P-LPS and LPS groups were also analyzed (Fig 7B and Table 6). A total of 24 metabolic pathways were

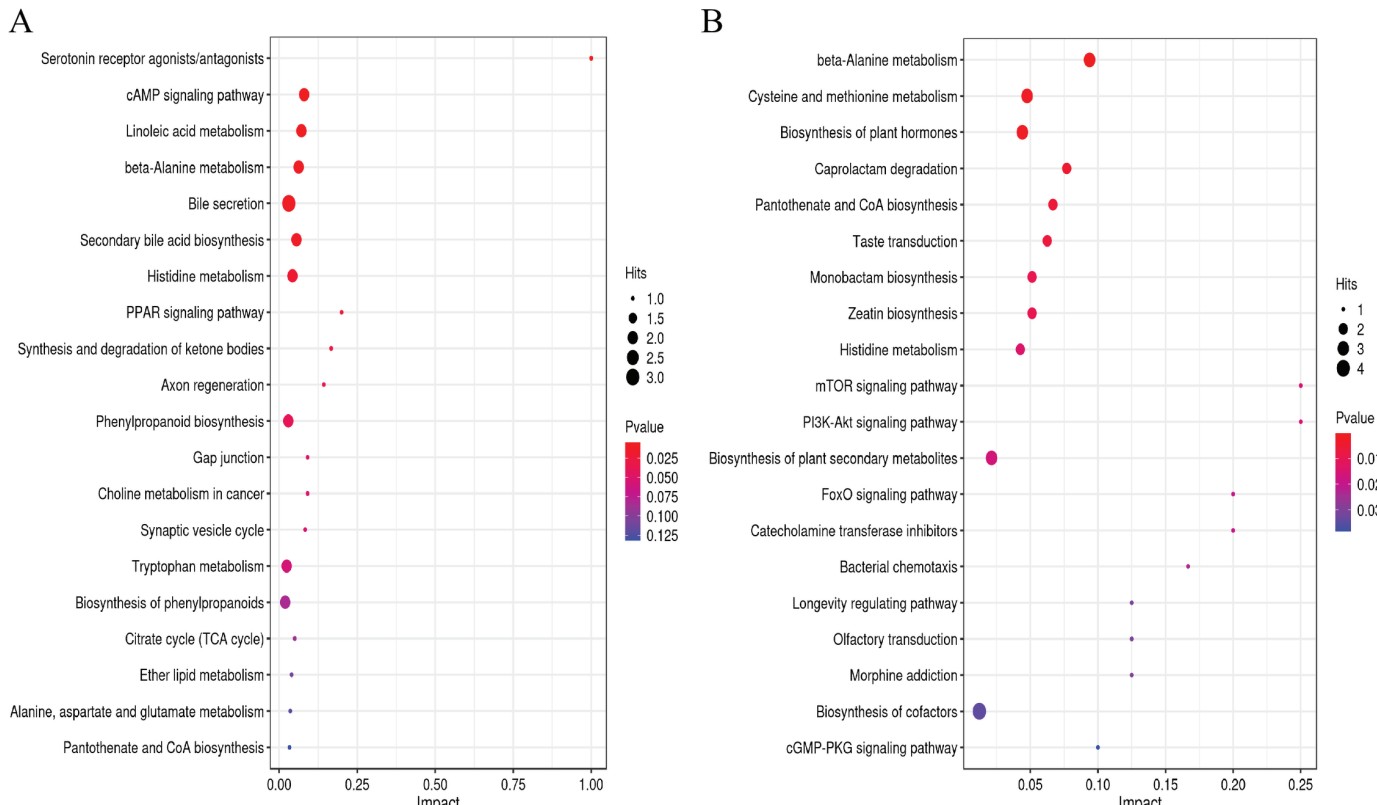

**Fig 7. The bubble plot shows the results of the Kyoto Encyclopedia of Genes and Genomes (KEGG) enrichment analysis of differential metabolites in positive and negative ion modes** ($n$ = 6). (A) LPS vs control; (B) P-LPS vs. LPS. Each point represents a metabolic pathway, with the x-axis indicating the impact value of the enriched pathways and the y-axis denoting the pathways. The size of the circles corresponds to the number of metabolites associated with each pathway. The color represents the p-value, where red indicates a smaller p-value (higher significance), and blue indicates a larger p-value (lower significance).

**Table 6. The KEGG enrichment analysis of the differential metabolites in serum between the P-LPS and LPS groups.**

| Map | Pathway | *p*-value | Compound Name |
|---|---|---|---|
| map00410 | beta-Alanine metabolism | 0.00023633 | L-Aspartic acid; Dihydrouracil; Anserine |
| map00270 | Cysteine and methionine metabolism | 0.00174855 | S-Adenosylmethionine; L-Aspartic acid; 3-Methylthiopropionic acid |
| map01070 | Biosynthesis of plant hormones | 0.00217883 | S-Adenosylmethionine; AMP; L-Aspartic acid |
| map00930 | Caprolactam degradation | 0.0044283 | 6-Hydroxyhexanoic acid; Epsilon-caprolactam |
| map00770 | Pantothenate and CoA biosynthesis | 0.00587135 | L-Aspartic acid; Dihydrouracil |
| map04742 | Taste transduction | 0.00666313 | AMP; Sucrose |
| map00261 | Monobactam biosynthesis | 0.00979131 | S-Adenosylmethionine; L-Aspartic acid |
| map00908 | Zeatin biosynthesis | 0.00979131 | S-Adenosylmethionine; AMP |
| map00340 | Histidine metabolism | 0.01401717 | L-Aspartic acid; Anserine |
| map04150 | mTOR signaling pathway | 0.01543007 | AMP |
| map04151 | PI3K-Akt signaling pathway | 0.01543007 | AMP |
| map01060 | Biosynthesis of plant secondary metabolites | 0.01652036 | S-Adenosylmethionine; AMP; L-Aspartic acid |
| map04068 | FoxO signaling pathway | 0.01925179 | AMP |
| map07216 | Catecholamine transferase inhibitors | 0.01925179 | 1,2,3-Trihydroxybenzene |
| map02030 | Bacterial chemotaxis | 0.02305929 | L-Aspartic acid |
| map04211 | Longevity regulating pathway | 0.03063183 | AMP |
| map04740 | Olfactory transduction | 0.03063183 | AMP |
| map05032 | Morphine addiction | 0.03063183 | AMP |
| map01240 | Biosynthesis of cofactors | 0.03419904 | S-Adenosylmethionine; AMP; L-Aspartic acid; Dehypoxanthine futalosine |
| map04022 | cGMP-PKG signaling pathway | 0.03814811 | AMP |

**Note:** Map ID = Enriched KEGG pathway ID; Map Title = Name of the enriched KEGG pathway; Compound_name = Name of the metabolites enriched in the pathway.

significantly affected ($P < 0.05$), with notable pathways being the mTOR signaling pathway, PI3K-Akt signaling pathway, and FoxO signaling pathway. Specifically, 4 metabolites were associated with the mTOR signaling pathway, 4 with the PI3K-Akt signaling pathway, and 5 with the FoxO signaling pathway. These results suggested that PAMK may mitigate the toxic effects of LPS on gosling intestines via the mTOR signaling pathway, PI3K-Akt signaling pathway, FoxO signaling pathway, and other related pathways. Based on the analysis of differential metabolic pathways between the P-LPS and LPS groups, PAMK was found to alleviate LPS-induced toxicity in gosling intestines by regulating the mTOR, PI3K-Akt, and FoxO signaling pathways.

**3.4.5 Associations between Gut microbiota and differential metabolites.** Spearman Rank Correlation was performed between 182 genera and 24 differential metabolites to determine the relationship between the abundance of differential metabolites and gut microbiota in the P-LPS vs LPS groups (Fig 8A). A total of 109 genera were significantly correlated with 24 differential metabolites, among which 9 metabolites were associated with regulation of the gut microbiota and had an impact on gut health or indirectly affected the composition and function of the gut microbiota. These metabolites included AMP, S-adenosylmethionine, erucic acid, sucrose, 3-methylxanthine, itaconic acid, L-aspartic acid, capric acid, and

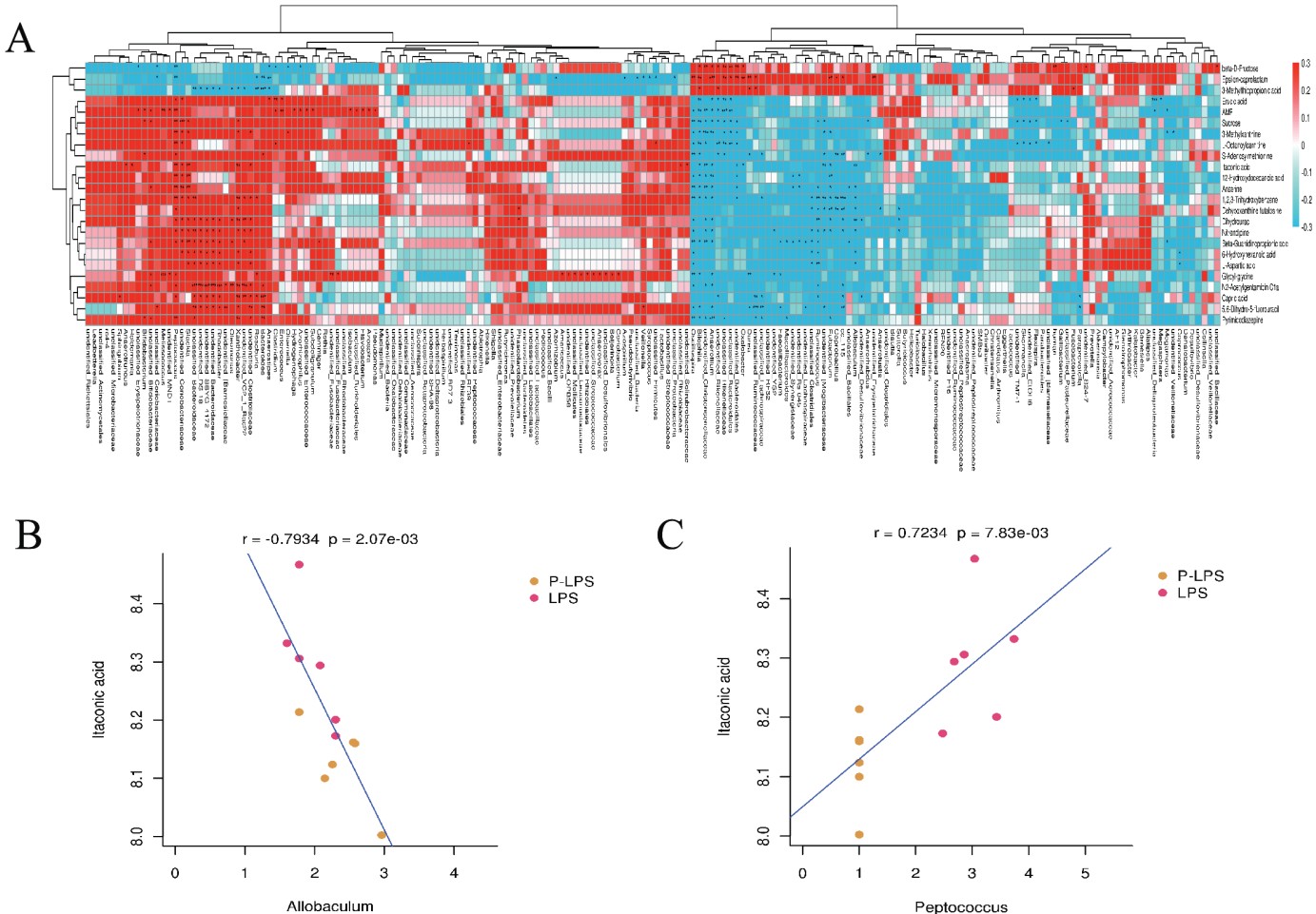

**Fig 8. Associations between Gut Microbiota and Differential Metabolites** (*n* = 6). (A) Spearman rank correlation between 24 differential metabolites and 182 gut microbes in the P-LPS group vs. PLS group. Each square represents the correlation and significance of this ID with the metabolite in other omics. Red represents positive correlation, blue represents negative correlation.The darker the color, the higher the correlation; the lighter the color, the lower the correlation. Statistical significance is indicated by stars—0.01 < *P* < 0.05: *; 0.001 < *P* < 0.01: **; *P* < 0.001: ***.(B and C) Scatter plot indicates the Person's correlation coefficient with statistical significance (*P* < 0.05) between Allobaculum or Peptococcus and serum Itaconic acid.

glycyl-glycine. Itaconic acid, a metabolite with anti-inflammatory properties, showed significant correlations with 17 genera, including *Allobaculum*, *Oxalobacter*, *Anaerofilum*, *Bilophila*, *Anaerotruncus* and *Peptococcus*. L Aspartic acid was significantly correlated with 13 genera, including *Coprococcus*, *Allobaculum*, *Ruminococcus*, *Oscillospira*, *Rhodobacter*, and *unidentified_Bacteroidaceae*. *Peptococcus* and *Allobaculum* both belong to the Peptococcaceae family within the Firmicutes phylum. Furthermore, the scatter plot used to visually highlight the correlation between itaconic acid and *Peptococcus* or *Allobaculum* (Fig 8B and 8C) demonstrated a positive correlation between the abundance of *Peptococcus* and *Allobaculum* and the levels of itaconic acid.

## 4 Discussion

Herbal medicine, as a natural functional feed additive, has gained increasing attention in recent years. These medicines can regulate the structure and metabolic function of the gut

microbiota, maintain its stability, and improve host immune function [19]. CRP and PCT are markers of systemic inflammation [20] [21]. LPS is known to increase the levels of inflammatory markers such as CRP in the blood of pigs, decrease physical activity and feed intake, and reduce norepinephrine levels in the brain [22]. Xiong et al. found that LPS could increase the permeability of intestinal epithelial cells and stimulate inflammatory cytokine production, leading to intestinal inflammation [23]. Wang et al. have reported that LPS-induced intestinal inflammation elevates the serum levels and mRNA expression of *HMGB*1, *TNF-α*, and *IL-*10 [24]. Xing et al. demonstrated that AOP could significantly suppress LPS-induced intestinal inflammatory responses by reducing the excessive production of inflammatory factors, including IL-1$\beta$ and IL-6, in the intestine [25]. Another study has suggested that PAMK could significantly alleviate dextran sodium sulfate-induced intestinal mucosal damage in mice by upregulating the expression of *Mucin-*2 and *Claudin-*1, enhancing the chemical barrier and tight junctions of the intestine, reducing the excessive expression of inflammatory cytokines (IL-1$\beta$, IL-6, and TNF-$\alpha$), and inhibiting neutrophil infiltration [26]. YuPingFengSan alleviates LPS-induced lung injury by suppressing the production of IL-1$\beta$, IL-6, and TNF-$\alpha$, inhibiting the activation of the NLRP3 inflammasome and MAPK signaling pathway, and improving intestinal barrier integrity while reducing gut inflammation [27]. Chen et al. have reported that PAMK reduces IL-1$\beta$ and IL-18 levels in LPS-induced inflammation in gosling livers [28]. Feng et al. have found that quercetin can upregulate the expression of genes such as *Claudin-*1 and *Mucin-*2, maintain intestinal epithelial cell integrity, and regulate intestinal inflammation [29]. Studies have also demonstrated that PAMK inhibits inflammatory responses, improves intestinal barrier integrity, reduces the activity of myeloperoxidase, a marker of neutrophil infiltration in colonic tissues, and decreases the expression of TNF-$\alpha$, IL-1$\beta$, IL-18, and IL-23. Moreover, it increases the expression of the tight junction proteins *ZO-*1 and *Occludin* [30]. These findings are consistent with the results from the current study, suggesting that PAMK alleviates LPS-induced intestinal injury in goslings and protects the intestinal mucosa.

TCM is characterized by its complex composition, which may include active compounds such as flavonoids, alkaloids, saponins, terpenes, and polysaccharides. These components are biologically active and considered the main therapeutic agents in TCM. Studies have reported that the gut microbiota can metabolize TCM polysaccharides to yield novel bioactive compounds, thereby influencing the therapeutic efficacy of TCM [31]. Researchers have demonstrated that Scutellaria baicalensis polysaccharide SP2-1 modulates the diversity and community structure of the gut microbiota by enhancing the abundance of beneficial bacteria such as *Bifidobacterium*, *Lactobacillus*, and *Roseburia* while inhibiting the proliferation of harmful bacteria including *Bacteroides* and *Staphylococcus* [32]. Gelsemium jasminoides Ellis polysaccharide can significantly alleviate gut microbiota dysbiosis in mice with cholestasis by increasing the abundance of *Enterococcaceae*, *Proteobacteria*, and *Firmicutes* while reducing that of *Bacteroidetes* and *Bacillota* [33]. Another study has reported that almond polysaccharide AP-1 can regulate the composition and abundance of gut microbiota, promote the growth of beneficial bacteria, suppress the proliferation of pro-inflammatory bacteria, and downregulate the mRNA expression of TNF-$\alpha$, IL-1$\beta$, IL-6, and inducible nitric oxide synthase in an LPS-induced model of inflammation [34]. Another study has discovered that PCP not only enhances the abundance of gut microbiota but also improves the $\alpha$ and $\beta$ diversities of the gut microbiota in mice with antibiotic-associated diarrhea [35]. Previous studies have shown that PAMK alleviates LPS-induced enteritis, maintains intestinal morphology, cytokine levels, tight junction integrity, and immunoglobulin levels, and improves gut microbiota dysbiosis in goslings [36]. Zhang et al. have demonstrated that PAMK can maintain the morphological integrity of the gut, improve gut microbiota dysbiosis, and alleviate LPS-induced enteritis

in goslings. PAMK may alleviate LPS-induced liver damage by regulating the p53 and FOXO signaling pathways [37]. Our findings revealed that the relative abundance of Proteobacteria decreased significantly and that of Firmicutes and Bacteroidetes increased significantly in the LPS group. In contrast, PAMK intervention decreased the abundance of Firmicutes and significantly increased the abundance of Bacteroidetes and Proteobacteria. It is speculated that LPS-induced inflammation triggers competitive energy scavenging and bacterial death among the gut microbiota, whereas PAMK alleviates intestinal inflammation and restores the energy balance, thereby modulating the microbial community structure. The increase in the abundance of Bacteroidetes in the LPS group might result from the competitive deprivation of nutrients. PAMK reduces this competitive behavior by stabilizing the microbial environment. Overall, PAMK alleviates LPS-induced gut dysbiosis by regulating the microbial abundance.

The effect of PAMK on serum metabolites was determined in LPS-treated goslings, a model characterized by increased levels of inflammatory cytokines and impaired intestinal barrier function. These changes further influence serum metabolite levels, including inflammation markers and compounds related to energy metabolism. Untargeted metabolomics is a technique used to analyze all detectable metabolites in a biological system rather than focusing on a predetermined set of target metabolites [38]. Multiple metabolites that exhibited significant changes following LPS treatment were identified using untargeted metabolomics. These metabolites are involved in various biological pathways. KEGG enrichment analysis of differential metabolites revealed changes in the levels of AMP, which plays a crucial role in energy transfer and metabolism. Fluctuations in AMP levels can activate or inhibit AMPK activation, thereby regulating energy metabolism. The dominant subunits of intestinal AMPK vary across species and are responsible for maintaining intestinal homeostasis and preventing intestinal diseases [39]. Studies have shown that the administration of Baitouweng decoction can repair the intestinal epithelial barrier in colitis by regulating the AMPK/mTOR mediated autophagy pathway [40]. Another study has reported that FebisGly can improve serum lipid metabolism in pigs, enhance the intestinal antioxidant capacity via the AMPK/FOXO pathway, and restore the gut microbiota and bile acid levels [41]. A study has shown that daidzein counteracts LPS-induced intestinal epithelial barrier damage, potentially via inhibition of the PI3K/AKT and P38 pathways [42]. Overall, PAMK may influence metabolic changes by regulating the mTOR, PI3K-Akt, and FoxO signaling pathways. These pathways play an essential role in cellular metabolism, inflammatory responses, and intestinal barrier function.

In this study, the combination of 16S rRNA sequencing and nontargeted metabolomics revealed the close relationship between metabolites and the gut microbiota. Spearman correlation analysis showed significant metabolic differences between the P-LPS and LPS groups, with the metabolites being closely correlated with changes in the abundance of the gut microbiota. Itaconic acid, an anti inflammatory metabolite, exhibited significant correlations with 17 genera including *Allobaculum*, *Peptococcus* and has been shown to alleviate intestinal inflammation by inhibiting inflammatory signaling pathways, including the NF-$\kappa$B pathway, thereby mitigating intestinal damage [43]. Both *Peptococcus* and *Allobaculum* produce short-chain fatty acids (SCFAs) in the gut, promoting intestinal health and maintaining the balance of the gut microbiota. SCFAs are crucial in regulating gut microbiota levels. Studies have reported the antihyperlipidemic effect of highland barley via its regulation of the gut microbiota and alteration of the abundance of SCFA producing bacteria in the gut [44]. SCFAs are important metabolites resulting from the fermentation of dietary fibers by gut microbes. They not only provide energy to intestinal epithelial cells but also participate in regulating the intestinal immune response and maintaining intestinal barrier function.

L-Aspartic acid showed significant correlations with 13 gut microbiota genera, including *Coprococcus* and *Ruminococcus*, suggesting that these amino acid metabolites may influence the metabolic balance and immune response of the gut by regulating the composition and function of the gut microbiota. L-Aspartic acid is closely related to intestinal barrier function and may maintain gut barrier integrity by regulating the function of intestinal epithelial cells [45]. Disruption of the intestinal barrier is a key factor in the onset of intestinal diseases. L-Aspartic acid may reduce inflammation and promote gut repair by affecting the function of both the intestinal barrier and immune cells. ZHTC can regulate the abundance of Firmicutes, Bacteroidetes, and other phyla in the gut microbiota, while altering the levels of 23 serum metabolites. Spearman correlation analysis has revealed significant correlations between metabolites such as arginine and specific microbiota [46]. In this study, a similar mechanism may also be at play, where changes in metabolites guide adaptive shifts in the abundance of gut microbiota.

Overall, our findings demonstrate the interplay between the gut microbiota and metabolites. Metabolites regulate the immune, barrier, and metabolic functions of the gut by influencing the composition and functionality of the gut microbiota. The study of these intricate relationships can offer new insights into the mechanisms of intestinal damage and repair. Future research could be designed to elucidate the functions of these metabolites in different pathological states of the gut and develop therapeutic strategies based on metabolite regulation of the gut microbiota.

## 5 Conclusion

PAMK exerted significant protective effects against LPS-induced intestinal injury in goslings. A total of 24 differential metabolites including L-aspartic acid and S-adenosylmethionine, which are involved in various metabolic pathways, were identified. Signal transduction pathways such as the mTOR, PI3K-Akt, and FoxO pathways were significantly affected by PAMK. PAMK could mitigate the toxic effects of LPS on the intestinal tract of goslings by modulating these metabolic pathways. PAMK could rectify the LPS-induced disruption of the intestinal microbiota composition and diversity. PAMK altered the abundance of Firmicutes, Bacteroidetes, and Proteobacteria at the phylum level. It influenced the abundance of the genera *Bacteroides*, *Lactobacillus*, *Faecalibacterium*, and also altered the dominant bacterial species in different groups. These findings provide a theoretical basis for the use of PAMK in maintaining the intestinal health of poultry and preventing intestinal diseases.

## Acknowledgements

I sincerely appreciate the contributions of Danning Xu and Yunbo Tian to the conception and design of this study. Shirou Pan and Shuaifei Bai made great efforts in data collection. Special thanks go to Jiayu He and Yunmao Huang for mainly conducting the animal experiments. Baili Lu and Xinliang Fu played crucial roles in data analysis. Yunmao Huang, Bingxin Li, and Nan Cao are to be thanked for the funding acquisition. Moreover, all authors participated in manuscript writing. All authors have read and agreed to the published version of the manuscript.

In addition, we are extremely grateful to the Guangdong Aquatic Poultry Science and Technology Innovation Platform for providing us with the experimental platform and necessary facilities. This support has been instrumental in enabling us to carry out various aspects of the research, including data collection, analysis, and interpretation, as well as cover expenses related to research materials and participant recruitment. Their investment in our work has significantly contributed to the quality and impact of our research findings.

## Author contributions

**Conceptualization:** Shirou Pan, Yunbo Tian.

**Data curation:** Shirou Pan, Shuaifei Bai, Junhao Wei, Ying Chen.

**Funding acquisition:** Bingxin Li, Nan Cao.

**Investigation:** Bingxin Li, Yunbo Tian.

**Methodology:** Shuaifei Bai, Yunbo Tian.

**Resources:** Jiayu He, Yunmao Huang.

**Software:** Jiayu He.

**Supervision:** Nan Cao, Yunmao Huang, Danning Xu.

**Validation:** Xinliang Fu, Danning Xu.

**Visualization:** Xinliang Fu, Danning Xu.

**Writing – original draft:** Baili Lu.

**Writing – review & editing:** Wanyan Li.

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
