## [Decision Letter · Decision Letter 0]

30 May 2025

PONE-D-25-01963Protection of LPS-Induced Intestinal Injury in Goslings by Polysaccharide of Atractylodes macrocephala Koidz Based on 16S rRNA and Metabolomics AnalysisPLOS ONE

Dear Dr. Xu,

Thank you for submitting your manuscript to PLOS ONE. After careful consideration, we feel that it has merit but does not fully meet PLOS ONE’s publication criteria as it currently stands. Therefore, we invite you to submit a revised version of the manuscript that addresses the points raised during the review process.

The reviewers have given sincere suggestions, please revise the manuscript according to the reviewer's suggestions.

We look forward to receiving your revised manuscript.

Kind regards,

Hongxun Tao

Academic Editor

PLOS ONE

Journal Requirements:

“the National Natural Science Foundation, grant number 32102747 and 32202764; the Science Technology Planning Project of Guangzhou, grant number 2023A04J0741 and 2023E04J0022, the Special Projects in Key Areas of General Universities in Guangdong Province  , grant number 2022ZDZX4022.”

“In addition, this study was supported by the following funds: the National Natural Science Foundation, grant numbers 32102747 and 32202764; the Science Technology Planning Project of Guangzhou, grant numbers 2023A04J0741 and 2023E04J0022; and the Special Projects in Key Areas of General Universities in Guangdong Province, grant number 2022ZDZX4022. The funding from these grants has provided important financial support for our research, enabling us to purchase necessary experimental materials and equipment, ensuring the smooth progress of the study.”

“the National Natural Science Foundation, grant number 32102747 and 32202764; the Science Technology Planning Project of Guangzhou, grant number 2023A04J0741 and 2023E04J0022, the Special Projects in Key Areas of General Universities in Guangdong Province  , grant number 2022ZDZX4022.”

5. We note that your Data Availability Statement is currently as follows: All relevant data are within the manuscript and its Supporting Information files.

**Additional Editor Comments:**

The reviewers have given sincere suggestions, please revise the manuscript according to the reviewer's suggestions.

Reviewers' comments:

Reviewer's Responses to Questions

**Comments to the Author**

1. Is the manuscript technically sound, and do the data support the conclusions?

Reviewer #1: Yes

Reviewer #2: Yes

Reviewer #3: Yes

2. Has the statistical analysis been performed appropriately and rigorously? 

Reviewer #1: Yes

Reviewer #2: Yes

Reviewer #3: Yes

3. Have the authors made all data underlying the findings in their manuscript fully available?

Reviewer #1: No

Reviewer #2: Yes

Reviewer #3: Yes

4. Is the manuscript presented in an intelligible fashion and written in standard English?

Reviewer #1: No

Reviewer #2: Yes

Reviewer #3: Yes

5. Review Comments to the Author

Reviewer #1: Reviewers' comments to editor:

This manuscript describes "Protection of LPS-Induced Intestinal Injury in Goslings by

Polysaccharide of Atractylodes macrocephala Koidz Based on 16S rRNA and Metabolomics Analysis".

The topic is of interest. However, there are several concerns about the study, and needs to be improved. Major revision.

1-In the abstract section, the sample types of the two omics need to be supplemented.

2- The important detection indicators should be listed in the abstract, and the conclusions need to be further clarified

3- In the introduction section, lack of epidemiological data support, please add it.

4-It is not clear why metabolomics technology is applied to this research direction, and relevant literature needs to be supplemented to support this research theme.

5-In the section of materials and methods, no clear ethical certification number for animal experiments is provided, please add.

6- No animal qualification number is provided

7- Detailed sample collection process and purpose? Sample sizes used for different experiments should be specifically described.

8- The item numbers and test steps of different kit indicators should be supplemented, as well as the detection limits.

9- The details of 16s sequencing sample collection need to be supplemented.

10- 16s sequencing pathways and functional analysis procedures need to be supplemented

11- Non-targeted metabolomics steps are too simple and important steps need to be described

12- The upstream and downstream direction of primers should be clearly marked, see Table 2

13- The statistical analysis method needs to be further improved, and the difference of methods should be properly explained for some normal and skew data

14-Indicators that appear for the first time should be given full English names

In the results section, it is necessary to supplement the mass spectrometry of key markers and compare them with standard substances.

15- The remarks below all the chart are not detailed.

16- Important indicators such as differential multiples of metabolites (FDR) and VIP need to be listed.

17- The dominant flora should be written in italics

18- It is suggested that the results should be analyzed jointly with the two omics,

19- It is suggested that a joint analysis of the results of the two omics should be made, and that the discussion should fully discuss how metabolites regulate the flora and affect the intestinal function, and supplement the relevant literature.

20- In conclusion, the important metabolites and metabolic pathways as well as the changes of dominant flora should be highlighted

21-In the result part, it is necessary to supplement the mass spectrometry of key markers, and identify and compare with standard substances. At the same time, provide the structural formula of key markers found, VIP value and other key identification information

22-In the discussion section, it is necessary to supplement the information about the new experimental results, and to explore the metabolites, metabolic pathways and potential pathogenic mechanisms in depth.

Other comments

1-It is suggested that references supplement the recent three years of research.

2-It is recommended that the article be polished by professionals or companies. There are currently problems with word writing or grammar.

Reviewer #2: This manuscript focuses on the protective effect of Polysaccharide of PAMK against LPS-induced intestinal injury in goslings, which is of great significance and high research value. The study centers on the practical problem in poultry farming where the intestinal health of goslings is threatened, affecting farming efficiency. By combining 16S rRNA and metabolomics analysis methods to explore the protective mechanism of PAMK, it is novel and innovative. In the experimental design, 1-day-old Magang goslings were randomly grouped, and the addition amount of PAMK and the injection dose of LPS were precisely controlled. A variety of scientific methods such as ELISA, 16S rRNA sequencing, non-targeted metabolomics technology, and real-time quantitative PCR were used to comprehensively and deeply explore its protective mechanism. The experiment obtained rich data covering multiple key aspects, and through reasonable statistical analysis, the results are reliable. In the discussion section, a wide range of literature was cited to deeply explore the protective mechanism of PAMK from multiple perspectives. The logic is rigorous, enhancing the persuasiveness of the research. Finally, a clear conclusion was drawn that PAMK has a protective effect on gosling intestinal injury, providing a potentially practical strategy for poultry farming with high application value. However, the synergistic mechanism between PAMK-regulated metabolic pathways and the improvement of the intestinal microbiota has not been deeply studied. Therefore, I believe that this manuscript needs to be revised. The specific comments are as follows:

Q1: How were the dosages of PAMK and LPS determined?

Q2: This study provides potential strategies for maintaining the intestinal health of poultry. However, the experiment was only conducted on goslings. In actual poultry farming production, the environmental factors are more complex, and different poultry may respond differently to PAMK. What further research needs to be carried out to apply the research results to actual farming?

Q3: As the key research object, what are the main components of PAMK?

Q4: The manuscript mentions that PAMK can protect the intestine by regulating the intestinal microbiota, enhancing intestinal barrier integrity, and regulating metabolic pathways, but the interrelationships among these three protective mechanisms have not been explored. For example, does the change in the intestinal microbiota affect the intestinal barrier function, and thus affect the metabolic pathways? Or do the changes in the metabolic pathways feedback - regulate the intestinal microbiota and the intestinal barrier?

Q5: When detecting serum inflammatory cytokines, only IL-1β, IL-6, IL-18, TNF-α, PCT, and CRP were measured. The intestinal inflammatory response involves a complex network of multiple cytokines. Is it necessary to detect more cytokines, such as anti-inflammatory cytokines like IL-10 and IL-4, to more comprehensively assess the balance of the inflammatory response?

Q6: In the results of 16S, why did the abundance of some beneficial bacteria genera increase in the LPS group, while it decreased in the PAMK + LPS group? What are the causes?

Q7: The manuscript has deficiencies in format and needs further adjustment and optimization. For example, in line 109, table 4.

Reviewer #3: The author investigated the protective effects of Polysaccharide of Atractylodes macrocephala Koidz (PAMK) on lipopolysaccharide (LPS)-induced intestinal injury in goslings using 16S rRNA and metabolomics analysis. PAMK exhibits protective effects against LPS-induced intestinal injury in goslings. After reviewing this manuscript, there are some changes that need to be made in regards to this study.

In Animals Grouping and Treatments part, the diets of the PAMK and P-LPS groups were supplemented with 400 mg/kg PAMK. How to provided PAMK for the PAMK and P-LPS groups?

In reagents part, all vendor details, including company, city, and country, should be mentioned for chemicals, reagents, strains, etc. Please complete vendor details.

In 16S rRNA Sequencing part, DNA was extracted using the OMEGA Soil DNA Kit. Why use soil DNA Kit for cecal contents?

In table 2, GenBank Accession should be provided.

In table 2, Gene symbols should be italicized.

In table 3, ”Total” indicates the total number of differential metabolites identified in each group. But in the group ,there were not 373 differentially expressed metabolites?

Double check the table title in Table 3.

There is a space between “eac” and “h” in Fig 6. legend.

In Differential Metabolic Pathways Indicating PAMK’s Role in Alleviating LPS-Induced Toxicity part, the authors only analyzed the differential metabolic pathways between the P-LPS group and the LPS group. The differential metabolic pathways of LPS group vs control group could be added.

Table 4. Primer Sequence?

6. PLOS authors have the option to publish the peer review history of their article (what does this mean?). If published, this will include your full peer review and any attached files.

Reviewer #1: **Yes: **Xinwen Dong

Reviewer #2: No

Reviewer #3: No

---

## [Author Response · Author response to Decision Letter 1]

15 Apr 2025

The comments have been carefully considered, and we have made detailed revisions. We appreciate your time and patience.

---

## [Decision Letter · Decision Letter 1]

27 May 2025

Protection of LPS-Induced Intestinal Injury in Goslings by Polysaccharide of Atractylodes macrocephala Koidz Based on 16S rRNA and Metabolomics Analysis

PONE-D-25-01963R1

Dear Dr. Xu,

We’re pleased to inform you that your manuscript has been judged scientifically suitable for publication and will be formally accepted for publication once it meets all outstanding technical requirements.

Kind regards,

Hongxun Tao

Academic Editor

PLOS ONE

Additional Editor Comments (optional):

Reviewers' comments:

Reviewer's Responses to Questions

**Comments to the Author**

1. If the authors have adequately addressed your comments raised in a previous round of review and you feel that this manuscript is now acceptable for publication, you may indicate that here to bypass the “Comments to the Author” section, enter your conflict of interest statement in the “Confidential to Editor” section, and submit your "Accept" recommendation.

Reviewer #1: All comments have been addressed

Reviewer #2: All comments have been addressed

Reviewer #3: All comments have been addressed

2. Is the manuscript technically sound, and do the data support the conclusions?

Reviewer #1: Yes

Reviewer #2: Yes

Reviewer #3: Yes

3. Has the statistical analysis been performed appropriately and rigorously? 

Reviewer #1: Yes

Reviewer #2: Yes

Reviewer #3: Yes

4. Have the authors made all data underlying the findings in their manuscript fully available?

Reviewer #1: Yes

Reviewer #2: Yes

Reviewer #3: Yes

5. Is the manuscript presented in an intelligible fashion and written in standard English?

Reviewer #1: Yes

Reviewer #2: Yes

Reviewer #3: Yes

6. Review Comments to the Author

Reviewer #1: (No Response)

Reviewer #2: (No Response)

Reviewer #3: The author has made substantial revisions based on the comments, and it is suggested that the manuscript can be accepted for publication.

7. PLOS authors have the option to publish the peer review history of their article (what does this mean?). If published, this will include your full peer review and any attached files.

Reviewer #1: No

Reviewer #2: No

Reviewer #3: No

---

## [Editor Report · Acceptance letter]

PONE-D-25-01963R1

PLOS ONE

Dear Dr. Xu,

I'm pleased to inform you that your manuscript has been deemed suitable for publication in PLOS ONE. Congratulations! Your manuscript is now being handed over to our production team.

Kind regards,

on behalf of

Dr. Hongxun Tao

Academic Editor

PLOS ONE